# E-MCTS: Deep Exploration by Planning with Epistemic Uncertainty

## Abstract

Deep model-based reinforcement learning (MBRL) is responsible for many of the greatest achievements of reinforcement learning. At the core of two of the approaches responsible for those successes, Alpha/MuZero, is a modified version of the Monte-Carlo Tree Search (MCTS) planning algorithm, replacing components of MCTS with learned models (of value and/or environment dynamics). Dedicated deep exploration, however, is a remaining challenge of Alpha/MuZero and by extension MCTS-based methods with learned models. To overcome this challenge, we develop Epistemic-MCTS. E-MCTS extends MCTS with estimation and propagation of epistemic uncertainty, and leverages the propagated uncertainty for a novel deep exploration algorithm by explicitly planning to explore. We incorporate E-MCTS into variations of MCTS-based MBRL approaches with learned (MuZero) and provided (AlphaZero) dynamics models. We compare E-MCTS to non-planning based deep-exploration baselines and demonstrate that E-MCTS significantly outperforms them in the investigated deep exploration benchmark.

## 1 Introduction

Deep model-based reinforcement learning (MBRL) has shown tremendous achievements in recent years, from super-human performance in games (Silver et al., 2018; Schrittwieser et al., 2020), through outperforming human designers in tasks that previously relied on intricate human engineering (Mandhane et al., 2022), to the design of novel algorithms (Mankowitz et al., 2023; Fawzi et al., 2022). At the heart of two of the best-performing MBRL approaches, Alpha/MuZero (Silver et al., 2018; Schrittwieser et al., 2020, responsible among others for the successes listed above) is the Monte-Carlo Tree Search (MCTS) planning algorithm. A remaining challenge in MCTS-based MBRL algorithms (that extends to Alpha/MuZero) is dedicated *deep exploration*. Deep exploration refers to the ability to find interesting states or state-actions, irrespective of how far away they are from the current state of the agent. This provides the agent with the means to spend its environment-interaction resources efficiently to gather new information and is critical in sparse-reward domains, providing up to exponential increases in sample efficiency Osband et al. (2016).

Standard approaches for deep exploration in reinforcement learning (RL) rely on estimates of *epistemic uncertainty* to direct the agent to unexplored areas of the state-action space (Osband et al., 2013; 2018; Bellemare et al., 2016; O'Donoghue et al., 2018). In line with existing literature (Hüllermeier & Waegeman, 2021) we define epistemic uncertainty as uncertainty that is *reducible with additional observations*. Note that this is different from uncertainty that is reducible with more *planning* (computation), as in classic MCTS, where the agent plans ahead using only rollouts of the true dynamics and does therefore not require exploration in the environment, only in the model. AlphaZero (Silver et al., 2018) replaces those rollouts with value and policy estimates, which are learned with neural networks and require sufficient exploration of the state-action space to be accurate. MuZero (Schrittwieser et al., 2020) additionally learns the transition and reward models from interactions, allowing it to solve model-free environments, but also increasing the demand for exploration to observe promising rewards in the environment. Our objective is to incorporate epistemic uncertainty into MCTS with learned value/dynamics models both to enable native deep exploration with MCTS-based algorithms, as well as to harness the strengths of MCTS for exploration, in the same way they are harnessed for exploitation.

In this work, we develop methodology to 1) incorporate epistemic uncertainty into MCTS, enabling agents to estimate the epistemic uncertainty associated with predictions at the root of the MCTS

planning tree (**E**pistemic-**MCTS**) and 2) leverage the uncertainty for deep exploration that capitalizes on the strengths of planning, by modifying the MCTS objective to an exploratory objective. We evaluate our agent on the benchmark hard-exploration task Deep Sea (Osband et al., 2020) against exploration baselines that do not leverage planning with uncertainty. In our experiments, our agent demonstrates deep exploration that significantly outperforms both naive and deep exploration baselines. The remainder of this paper is organized as follows: Section 2 provides relevant background for MBRL, MCTS and epistemic uncertainty estimation in deep RL. Section 3 describes our contributions, starting with distinguishing between epistemic and non-epistemic sources in MCTS, followed by the framework for uncertainty propagation in MCTS (E-MCTS), our approach for harnessing E-MCTS to achieve deep exploration and finally a discussion regarding the challenges and limitations in estimating epistemic uncertainty in planning with an abstracted, learned model of the environment. Section 4 discusses related work. Section 5 evaluates our method with different dynamics models against a hard-exploration benchmark and compares to standard exploration baselines. Finally, Section 6 concludes the paper and discusses future work.

## 2 BACKGROUND

In RL, an agent learns a behavior policy $\pi(a|s)$ through interactions with an environment, by observing states (or observations), executing actions and receiving rewards. The environment is represented by a Markov Decision Process (MDP, Bellman, 1957), or a partially-observable MDP (POMDP, Åström, 1965). An MDP $\mathcal{M}$ is a tuple: $\mathcal{M} = \langle \mathcal{S}, \mathcal{A}, \rho, P, R \rangle$, where $\mathcal{S}$ is a set of states, $\mathcal{A}$ a set of actions, $\rho$ the initial state distribution, $R : \mathcal{S} \times \mathcal{A} \times \mathcal{S} \to \mathbb{R}$ a bounded reward function, and $P : \mathcal{S} \times \mathcal{A} \times \mathcal{S} \to [0,1]$ is a transition function, where $P(s_{t+1}|s_t, a_t)$ specifies the probability of transitioning from state $s_t$ to state $s_{t+1}$ after executing action $a_t$ at time $t$. In a POMDP $\mathcal{M}' = \langle \mathcal{S}, \mathcal{A}, \rho, P, R, \Omega, O \rangle$, the agent observes observations $o_t \in \Omega$. $O : \mathcal{S} \times \mathcal{A} \times \Omega \to [0,1]$ specifies the probability $O(o|s_t, a_t)$ of observing $o$. In MBRL the agent uses a model of the environment to optimize its policy, often through planning. The model is either learned from interactions, or provided. In Deep MBRL (DMBRL) the agent utilizes deep neural networks as function approximators. Many RL approaches rely on learning a state-action *Q-value function* $Q^\pi(s, a) = \mathbb{E}[R(s, a, s') + \gamma V^\pi(s') \,|\, s' \sim P(\cdot|s,a)]$ or the corresponding state *value function* $V^\pi(s) = \mathbb{E}[Q^\pi(s, a) \,|\, a \sim \pi(\cdot|s)]$, which represents the expected return from starting in state $s$ (and possibly action $a$) and then following a policy $\pi(a_t|s_t)$ which specifies the probability of selecting the action $a_t$ in state $s_t$. The discount factor $0 < \gamma < 1$ is used in infinite-horizon (PO)MDPs to guarantee that the values remain bounded, and is commonly used in RL for learning stability.

### 2.1 MONTE CARLO TREE SEARCH

MCTS is a planning algorithm that constructs a planning tree with the current state $s_t$ at its root to estimate the objective: $\arg\max_a \max_\pi Q^\pi(s_t, a)$. The algorithm iteratively performs *trajectory selection*, *expansion*, *simulation* and *backup* to arrive at better estimates at the root of the tree. At each planning step $i$, starting from the root node $s_{t,0}^i \equiv \hat{s}_0$, the algorithm selects a trajectory in the existing tree based on the averaged returns $q(\hat{s}_k, a)$ experienced in past trajectories selecting the action $a$ in the same node $\hat{s}_k$, and a search heuristic, such as an Upper Confidence Bound for Trees (UCT, Kocsis & Szepesvári, 2006):

$$a_k = \arg\max_{a \in A} q(\hat{s}_k, a) + C \sqrt{\frac{2\log(\sum_{a'} N(\hat{s}_k, a'))}{N(\hat{s}_k, a)}}, \tag{1}$$

where $N(\hat{s}_k, a)$ denotes the number of times action $a$ has been executed in node $\hat{s}_k$, and $C > 0$ trades off exploration of new nodes with maximizing observed return. When the the trajectory selection arrives at a leaf node $\hat{s}_T$ MCTS expands the node and estimates its initial value as the average of Monte-Carlo rollouts using a random policy. Recent DMBRL algorithms that use MCTS such as Alpha/MuZero (Silver et al., 2016; 2017; 2018; Schrittwieser et al., 2020) replace the rollouts with a value function $v(\hat{s}_T)$ that is approximated by a neural network and use the PUCT (Rosin, 2011) search heuristic instead of UCT:

$$a_k = \arg\max_{a \in A} q(\hat{s}_k, a) + \pi(a|\hat{s}_k) C \frac{\sqrt{\sum_{a'} N(\hat{s}_k, a')}}{1 + N(\hat{s}_k, a)}. \tag{2}$$

Where $\pi(a|\hat{s}_k)$ is either given, or learned by imitating the MCTS policy $\pi^{\text{MCTS}}$, to incorporate prior knowledge into the search. MCTS propagates the return (discounted reward for visited nodes plus

leaf's value) back along the planning trajectory. At the root of the tree, the optimal value $\max_\pi V^\pi(s_t)$ of current state $s_t$ is estimated based on the averaged returns experienced through every action $a$, and averaged over the actions:

$$\max_\pi V^\pi(s_t) \approx \sum_{a \in \mathcal{A}} \frac{N(\hat{s}_0, a)}{\sum_{a' \in \mathcal{A}} N(\hat{s}_0, a')} q(\hat{s}_0, a) =: \sum_{a \in \mathcal{A}} \pi^{\text{MCTS}}(a|s_t) q(\hat{s}_0, a) =: v_t^{\text{MCTS}}. \quad (3)$$

## 2.2 MCTS-Based Model Based Reinforcement Learning

MCTS requires access to three core functions. Those are: (i) a representation function $g(s_t) = \hat{s}_0 \in \hat{\mathcal{S}}$ that encodes the current state at the root of the tree into a latent space, in which (ii) a transition function $f(\hat{s}_k, a_k) = \hat{s}_{k+1}$ predicts the next latent state and (iii) a function $r(\hat{s}_k, a_k) = \mathbb{E}[r_k|\hat{s}_k, a_k]$ that predicts the corresponding average reward. Such models in an latent state space $\hat{\mathcal{S}} \neq \mathcal{S}$ do not have to distinguish between different true states $s, s' \in \mathcal{S}$, i.e., $g(s) = g(s'), s \neq s'$, if such a distinction does not benefit value and reward prediction, and are commonly called *value-equivalent* or *abstracted* models. Note that for an identity function $g(s_t) = s_t$ all models, functions and policies would be defined in the true state space $\mathcal{S}$, and that in a POMDP $g$ can encode the current observation $o_t$ or the entire action-observation history $\langle o_0, a_0, o_1, a_1, \ldots, o_t \rangle$. As in Mu/AlphaZero (Schrittwieser et al., 2020; Silver et al., 2018), a value function $v(\hat{s}_T)$ can be learned for replacing rollouts, and a policy function $\pi(a|\hat{s}_k)$ imitates the MCTS policy to bias planning towards promising actions based on prior knowledge. In deep MBRL (DMBRL) these functions are learned with deep neural networks. Five common learning signals are used to train the transition model $f$ with varying horizons $k$:

1) A reconstruction loss $L_{re}^k(h(\hat{s}_k), s_{t+k})$, training a decoder $h$ to reconstruct true states $s_{t+k}$ from latent representations $\hat{s}_k$ that have been predicted from $\hat{s}_0 = g(s_t)$, shaping both $g$ and $f$.

2) A consistency loss $L_{co}^k(\hat{s}_k, g(s_{t+k}))$, training the model that predicted states should align with latent representation of states $s_t$ (or observations/histories in POMDP). Critically, $L_{co}^k$ is not used to train $g$, only $f$. When the representation function $g$ is an identity, $L_{re}^k$ and $L_{co}^k$ can be thought of as providing the same learning signal. Otherwise, they can be used independently or in combination.

3) A reward loss $L_r^k(r(\hat{s}_k, a_k), r_{t+k})$, where the model is trained to predict representations that enable predictions of, and are aligned with, the true rewards observed in the environment $r_t$.

4) A value loss $L_v^k(v(\hat{s}_k), v_{t+k}^{\text{MCTS}})$ that trains the model to predict states that enable value learning.

5) A policy loss $L_\pi^k(\pi(\cdot|\hat{s}_k), \pi^{\text{MCTS}}(\cdot|s_{t+k}))$ that trains prior policy $\pi$ to predict the MCTS policy.

These losses are described in more detail in Appendix D.2.

## 2.3 Estimating Epistemic Uncertainty in Deep Reinforcement Learning

Predictive epistemic uncertainty refers to any uncertainty that is associated with a prediction and is rooted in lack-of-information. For example, prior to repeated tosses of a coin, there can be high uncertainty whether the coin is fair or not. The more the coin has been tossed, the more certain we can be about the coin's fairness, even if we will always retain uncertainty in the exact prediction of heads or tails, without access to a precise simulation of the physics of the coin toss (referred to as *aleatoric* uncertainty, or the inherent uncertainty in the way we choose to model a coin). Defining, quantifying and estimating predictive epistemic uncertainty is an active field of research that encompasses many approaches and many methods (see Hüllermeier & Waegeman, 2021; Lockwood & Si, 2022). In this work, we take the common approach for quantifying epistemic uncertainty as the variance in a probability distribution of predictions that are consistent with observations $\text{Var}_X(X|s_t) = \mathbb{V}[X|s_t]$.

As for estimating epistemic uncertainty, two standard approaches are the distributional approach and the proxy-based approach. The distributional approach approximates a probability distribution over possible predictions with respect to the agent's experiences, while the proxy-based approach aims to directly predict a measure for *novelty* of experiences. Two reliable and lightweight methods for novelty-based epistemic uncertainty estimation are Random Network Distillation (RND) (Burda et al., 2019) and state-visitation counting. RND evaluates novelty as the difference between the prediction of a randomly initialized untrained target network $\psi'$ and a to-be trained network $\psi$ with a similar architecture. The network $\psi$ is trained to match the predictions of the target network for the observed states (or state-action pairs) with MSE loss $L_{rnd}(\psi(s_t, a_t), \psi'(s_t, a_t)) = ||\psi(s_t, a_t) - \psi'(s_t, a_t)||^2$. Novel observations are expected to produce unpredictable outputs from the target network, and thus the difference between the prediction of the target network and the trained network serves as a proxy-measure for novelty. These methods encapsulate the epistemic uncertainty in a local prediction:

for example, uncertainty in prediction of reward or next state. Estimating epistemic uncertainty in value predictions that contain the uncertainty that propagates from future decisions made by a policy is a different matter. One method to estimate value uncertainty is the Uncertainty Bellman Equation (UBE, O'Donoghue et al., 2018). UBE approximates an upper bound on the epistemic uncertainty in value (here interpreted as variance of the Q-value) as the sum of local uncertainties $\sigma^2(s_t, a_t)$ that are associated with the decisions $a_t$ at states $s_t$:

$$U^\pi(s_t) \;:=\; \mathbb{E}_\pi\left[\sum_{i=0}^{\infty}\gamma^{2i}\sigma^2(s_{t+i}, a_{t+i}^\pi)\right] \;=\; \mathbb{E}_\pi\left[\sum_{i=0}^{n-1}\gamma^{2i}\sigma^2(s_{t+i}, a_{t+i}^\pi) + \gamma^{2n}U^\pi(s_{t+n})\right].$$

In other words, UBE proposes to approximate the value uncertainty as the sum of twice-discounted local uncertainties and learn it with (possibly $n$-step) TD targets in a similar manner to value learning.

## 3 DEEP EXPLORATION WITH EPISTEMIC MCTS

In this work we are concerned with estimating and leveraging epistemic uncertainty in MCTS to drive exploration in the environment. In classic MCTS, the uncertainty in value prediction at each node stems from stochasticity in the environment and in the rollout policy (aleatoric). There are no learned quantities, and as such, there is no epistemic uncertainty in the model used by MCTS. When a learned value function $v(s_t)$ is used to replace rollouts (such as in AlphaZero, Silver et al., 2018) the aleatoric uncertainty from MC rollouts is replaced by uncertainty in the value prediction $v(s_t)$. We distinguish between two sources of uncertainty about $v(s_t)$: 1. Epistemic sources: errors resulting from evaluating $v(s_t)$ on unobserved states $s_t$. 2. Non-epistemic sources: approximation errors, TD-errors, stochasticity of the environment and the policy and every other source of error that will not reduce directly by training on additional unique observations. When a model of the environment transition $f(s_t, a_t)$ and/or reward dynamics $r(s_t, a_t)$ are learned from interactions (such as in MuZero, Schrittwieser et al., 2020) the uncertainty in value of a node in the planning tree will contain the uncertainty in the learned dynamics $f, r$, and a similar separation between epistemic and non-epistemic sources of uncertainty can be made.

MCTS addresses non-epistemic uncertainty by averaging over node values and using UCB-based exploration in the planning tree, but does not address epistemic uncertainty in the model (there isn't any, in classic MCTS). Unlike the uncertainty estimated by the UCB bonus of MCTS, epistemic uncertainty cannot be expected to reduce directly as a result of additional planning: rather, epistemic uncertainty will only reduce as a result of 1. new interactions with the environment and 2. planning in directions where the agent is more epistemically-certain. Distinguishing between epistemic and non-epistemic uncertainty allows us to concentrate on propagating only epistemic uncertainty for exploration. In the following section we develop a method to propagate the epistemic uncertainty in MCTS (Section 3.1). We follow by leveraging the propagated uncertainty into an exploratory epistemic-UCB planning objective (Section 3.2). To conclude this section, we discuss challenges in estimating novelty when planning in latent spaces and possible solutions (Section 3.3).

### 3.1 PROPAGATING UNCERTAINTY IN MCTS

At planning step $i$, selecting a path of length $T$ through a decision tree is equivalent to choosing a sequence of $T$ actions $a_{0:T-1}^i$ that start at node $\hat{s}_0^i = g(s_t)$ and end up in a leaf node $\hat{s}_T^i$. Deterministic models $f, r$ predict the transitioned to nodes $\hat{s}_k^i$ and the encountered rewards $r_k^i$ in nodes $\hat{s}_k^i, 0 \le k < T$, respectively. The value $v_T^i$ at leaf $\hat{s}_T^i$ is predicted by Monte-Carlo rollouts with $f$ or directly with a neural network $v$. The values and rewards are used to update the $n$-step discounted return $\nu_k^i$ of each node $\hat{s}_k^i$ on the selected path:

$$\nu_k^i \quad := \quad \sum_{j=k}^{T-1}\gamma^{j-k}r_j^i + \gamma^{T-k}v_T^i \quad = \quad r_k^i + \gamma\nu_{k+1}^i, \qquad 0 \le k < T, \qquad \nu_T^i = v_T^i, \quad (4)$$

where $\gamma^{j-k}$ is the discount factor to the power of $j - k$ and the superscript $i$ is indexing the planning step. Our following analysis is done per planning step $i$ and we will drop the index $i$ for the sake of readability. If (any of) $f, r, v$ are assumed to be inexact $r_k$ and $v_T$ can be modelled as random variables in a Markov chain that is connected by random state-variables. The stochasticity in the chain captures the uncertainty in $f, r, v$'s predictions. To clarify notation, we will refer to these as random states $\hat{S}_k$, rewards $R_k$, values $V_k$ and returns $\mathcal{V}_k$. In line with the optimistic exploration literature, we

aim to incentivize choosing actions in the environment associated with paths in the planning tree that have *epistemically* uncertain returns $\mathcal{V}_0$ in order to seek new high-reward interactions. For this we need to estimate the *epistemic* variance (variance from epistemic sources) $\mathbb{V}[\mathcal{V}_0|s_t, a_{0:T-1}] \equiv \mathbb{V}[\mathcal{V}_0]$ of the return along a selected path $a_{0:T-1}$, starting with state $s_t$. To circumvent having to replace $f, r, v$ with an explicitly stochastic model to propagate the uncertainty, we instead develop a direct and computationally efficient approximation for $\mathbb{V}[\mathcal{V}_0]$.

We will begin by deriving the mean and variance of the distribution of state-variables in the Markov chain for a given sequence of actions $a_{0:T-1}$. Let us assume we are given a differentiable transition function $f(\hat{S}_k, a_k) := \mathbb{E}_{\hat{S}_{k+1}}[\hat{S}_{k+1}|\hat{S}_k, a_k] \in \mathbb{R}^{|\hat{\mathcal{S}}|}$, which predicts the conditional expectation over the next state, and a differentiable uncertainty function $\boldsymbol{\Sigma}(\hat{S}_k, a_k) := \mathbb{V}_{\hat{S}_{k+1}}[\hat{S}_{k+1}|\hat{S}_k, a_k] \in \mathbb{R}^{|\hat{\mathcal{S}}| \times |\hat{\mathcal{S}}|}$ that yields the conditional-covariance matrix of the distribution. In DMBRL the assumption that models are differentiable is standard (see Section 2.2). We assume that the mean $\hat{s}_0$ of the first state-variable $\hat{S}_0$ is given as an encoding function $\hat{s}_0 = \mathbb{E}[\hat{S}_0|s_t] = g(s_t)$, like in MuZero. The mean $\hat{s}_{k+1}$ of a later state-variable $\hat{S}_{k+1}$ can be approximated with a first order Taylor expansion around the previous mean $\hat{s}_k := \mathbb{E}[\hat{S}_k]$:

$$\begin{aligned}
\hat{s}_{k+1} &:= \mathbb{E}[\hat{S}_{k+1}] = \mathbb{E}_{\hat{S}_k}[\mathbb{E}_{\hat{S}_{k+1}}[\hat{S}_{k+1}|\hat{S}_k, a_k]] = \mathbb{E}[f(\hat{S}_k, a_k)] \quad (5)\\
&\approx \mathbb{E}[f(\hat{s}_k, a_k) + (\hat{S}_k - \hat{s}_k)^\top \nabla_{\hat{s}} f(\hat{S}, a_k)|_{\hat{S}=\hat{s}_k}] = f(\hat{s}_k, a_k).
\end{aligned}$$

In other words, under the assumption that the model $f$ predicts the *expected* next state we reinterpret the original latent state $\hat{s}_k$ as the mean of the uncertain state $\mathbb{E}[\hat{S}_k]$.

To approximate the covariance $\boldsymbol{\Sigma}_{k+1} := \mathbb{V}[\hat{S}_{k+1}]$ or *the total uncertainty associated with state variable* $\hat{S}_{k+1}$ we need the *law of total variance*. The law of total variance states that for two random variables $X$ and $Y$ holds $\mathbb{V}[Y] = \mathbb{E}_X[\mathbb{V}_Y[Y|X]] + \mathbb{V}_X[\mathbb{E}_Y[Y|X]]$ (see Appendix A for a proof in our notation). Using the law of total variance and again a first order Taylor approximation around the previous mean state $\hat{s}_k$, where $\boldsymbol{J}_f(\hat{s}_k, a_k)$ denotes the Jacobian matrix of function $f$ at $\hat{s}_k$ and $a_k$:

$$\begin{aligned}
\boldsymbol{\Sigma}_{k+1} &:= \mathbb{V}[\hat{S}_{k+1}] = \underbrace{\mathbb{E}_{\hat{S}_k}[\mathbb{V}_{\hat{S}_{k+1}}[\hat{S}_{k+1}|\hat{S}_k, a_k]]}_{\boldsymbol{\Sigma}(\hat{s}_k, a_k)} + \underbrace{\mathbb{V}_{\hat{S}_k}[\mathbb{E}_{\hat{S}_{k+1}}[\hat{S}_{k+1}|\hat{S}_k, a_k]]}_{\boldsymbol{J}_f(\hat{s}_k, a_k)\, \boldsymbol{\Sigma}_k\, \boldsymbol{J}_f(\hat{s}_k, a_k)^\top} \quad (6)\\
&\approx \qquad\qquad \boldsymbol{\Sigma}(\hat{s}_k, a_k) \qquad\qquad + \qquad \boldsymbol{J}_f(\hat{s}_k, a_k)\, \boldsymbol{\Sigma}_k\, \boldsymbol{J}_f(\hat{s}_k, a_k)^\top.
\end{aligned}$$

See Appendix B for the full derivation. Using these state statistics, we can derive the means and variances of causally connected variables like rewards $R_k$ and values $V_T$. We assume that the conditional reward distribution has conditional mean $r(\hat{S}_k, a_k) := \mathbb{E}_{R_k}[R_k|\hat{S}_k, a_k]$ and conditional variance $\sigma_R^2(\hat{S}_k, a_k) := \mathbb{V}_{R_k}[R_k|\hat{S}_k, a_k]$, and that the conditional value distribution has conditional mean $v(\hat{S}_T) := \mathbb{E}_{V_T}[V_T|\hat{S}_T]$ and conditional variance $\sigma_V^2(\hat{S}_T) := \mathbb{V}_{V_T}[V_T|\hat{S}_T]$. Analogous to above we can derive:

$$r_k := \mathbb{E}[R_k] \approx r(\hat{s}_k, a_k), \qquad \mathbb{V}[R_k] \approx \sigma_R^2(\hat{s}_k, a_k) + \boldsymbol{J}_r(\hat{s}_k, a_k)\, \boldsymbol{\Sigma}_k\, \boldsymbol{J}_r(\hat{s}_k, a_k)^\top, \quad (7)$$

$$v_T := \mathbb{E}[V_T] \approx v(\hat{s}_T), \qquad \mathbb{V}[V_T] \approx \sigma_V^2(\hat{s}_T) + \boldsymbol{J}_v(\hat{s}_T)\, \boldsymbol{\Sigma}_T\, \boldsymbol{J}_v(\hat{s}_T)^\top. \quad (8)$$

If we assume that $R_k$ and the $n$-step return $\mathcal{V}_{k+1}$ from Equation 4 are independent, we can compute

$$\mathbb{E}[\mathcal{V}_k] = \mathbb{E}_{R_k, \mathcal{V}_{k+1}}[R_k + \gamma \mathcal{V}_{k+1}] = \mathbb{E}[R_k] + \gamma\, \mathbb{E}[\mathcal{V}_{k+1}], \qquad \mathbb{E}[\mathcal{V}_T] = \mathbb{E}[V_T], \quad (9)$$

$$\mathbb{V}[\mathcal{V}_k] = \mathbb{V}_{R_k, \mathcal{V}_{k+1}}[R_k + \gamma \mathcal{V}_{k+1}] = \mathbb{V}[R_k] + \gamma^2\, \mathbb{V}[\mathcal{V}_{k+1}], \quad \mathbb{V}[\mathcal{V}_T] = \mathbb{V}[V_T]. \quad (10)$$

We can therefore approximate the variance $\mathbb{V}[\mathcal{V}_0|s_t, a_{0:T-1}]$ using one (E-)MCTS search, expansion and back-propagation steps through the selected path $a_{0:T-1}$, similar to the value-estimation $\mathbb{E}[\mathcal{V}_0|s_t, a_{0:T-1}]$ that is being done by standard MCTS (see pseudo-code in Algorithm 1). When applying this approach to model-learning algorithms such as MuZero, we interpret the representation $g$, dynamics $f$, value $v$ and reward $r$ functions as outputting the conditional means $\hat{s}_0, \hat{s}_k, v_T, r_k$ respectively. When applying this approach to methods that learn only some of $f, r, v$ (for example AlphaZero, Silver et al., 2018, which learns only $v$) the predictions from unlearned components will be associated with epistemic uncertainty $= 0$. E-MCTS will propagate the epistemic uncertainty in the learned components according to the remaining nonzero terms in Equations 6, 7, 8, 10. Finally we note that while E-MCTS is designed with epistemic uncertainty of the learned models in mind, any source of uncertainty can be propagated with E-MCTS, so long as it is interpreted as the local variances in state, reward and value predictions (Equations 6, 7 and 8 respectively).

## 3.2 Planning for Exploration with MCTS

The UCT operator of MCTS takes into account uncertainty about a node's subtree via the visitation count (see Equation 1) to drive exploration *inside* the planning tree and identify the most promising expected-return-maximizing actions in the model. To drive exploration *in the environment* we add the environmental epistemic uncertainty to the selection step, which maximizes an upper confidence bound on the agent's knowledge of both the *environment* (in blue) and the *search tree* (the original UCT bonus):

$$a_k \quad := \quad \arg\max_a q(\hat{s}_k, a) + \beta\sqrt{\sigma_q^2(\hat{s}_k, a_k)} + C\sqrt{\frac{2\log(\sum_{a'} N(\hat{s}_k, a'))}{N(\hat{s}_k, a_k)}}, \tag{11}$$

where $\beta \geq 0$ is a constant that can be tuned per task to encourage more or less exploration in the environment. The term

$$\sigma_q^2(\hat{s}_k, a_k) \quad := \quad \mathbb{V}[R_k] + \gamma^2 \frac{1}{N(\hat{s}_k, a_k)}\sum_{i=1}^{N(\hat{s}_k, a_k)} \mathbb{V}[\mathcal{V}_{k+1}^i] \tag{12}$$

sums the variances computed individually at every backup step $i$ through the node that is reached by executing action $a_k$ in latent state $\hat{s}_k$ using Equations 7 and 10. At each backup step $i$, with actions $a_k^i$, state means $\hat{s}_k^i$ and covariances $\boldsymbol{\Sigma}_k^i$, the variance $\mathbb{V}[\mathcal{V}_k^i]$ is approximated based on Equations 10 and 7:

$$\mathbb{V}[\mathcal{V}_k^i] \approx \sigma_R^2(\hat{s}_k^i, a_k^i) + \boldsymbol{J}_r(\hat{s}_k^i, a_k^i)\boldsymbol{\Sigma}_k^i \boldsymbol{J}_r(\hat{s}_k^i, a_k^i)^\top + \gamma^2 \mathbb{V}[\mathcal{V}_{k+1}^i]. \tag{13}$$

At every backup step we compute the variance at the leaf node (Equation 8), which is then used to update the parent's variance along the trajectory iteratively using Equation 13. Pseudo-code can be found in Algorithm 1, where the modifications introduced to MCTS are marked in blue. When using other search heuristics such as PUCT or the extension of PUCT used in Gumbel MuZero (Danihelka et al., 2022) we propose to view the term $q(\hat{s}_k, a) + \beta\sqrt{\sigma_q^2(\hat{s}_k, a_k)}$ as an exploratory-Q-value-estimate (or epistemically-optimistic-Q-value estimate) and use it in place of $q(\hat{s}_k, a)$ to modify the planning objective into the exploratory objective. Once the MCTS-based search with respect to the exploratory Q-value has completed, action selection in the environment can be done in the same manner as for exploitation. For example, by sampling actions with respect to the visitation counts of each action at the root of the tree as done by the original MuZero.

---

**Algorithm 1** E-MCTS, requires functions $g, f, r, v$ and uncertainty estimators $\boldsymbol{\Sigma}, \sigma_R^2, \sigma_V^2$

1: **function** EMCTS(state $s_t$, $\beta$)         $\triangleright \beta = 0$ for unmodified MCTS exploitation episodes
2:     **while** within computation budget **do**
3:        SELECT($g(s_t)$, $\beta$)       $\triangleright$ traverses tree from root $\hat{s}_0 = g(s_t)$ and adds new leaf
4:     **return** action $a$ drawn from $\pi(a|s_t) = \frac{N(\hat{s}_0, a)}{\sum_{a'} N(\hat{s}_0, a')}$      $\triangleright$ MCTS action selection

5: **function** SELECT(node $\hat{s}_k$, $\beta$)
6:     $a_k \leftarrow \arg\max_a q(\hat{s}_k, a) + \beta\sqrt{\sigma_q^2(\hat{s}_k, a)} + C\sqrt{\frac{2\log(\sum_{a'} N(\hat{s}_k, a'))}{N(\hat{s}_k, a)}}$     $\triangleright$ Equation 11
7:     **if** $a_k$ already expanded **then** SELECT($f(\hat{s}_k, a_k)$, $\beta$)      $\triangleright$ traverses tree
8:     **else** EXPAND($\hat{s}_k, a_k$)      $\triangleright$ adds new leaf

9: **function** EXPAND(node $\hat{s}_k$, not yet expanded action $a_k$)
10:     $\hat{s}_{k+1}, \mathbb{E}[V_{k+1}] \leftarrow$ Execute unmodified MCTS expansion that creates a new leaf $\hat{s}_{k+1}$
11:     $\boldsymbol{\Sigma}_{k+1} \leftarrow \boldsymbol{\Sigma}(\hat{s}_k, a_k) + \boldsymbol{J}_f(\hat{s}_k, a_k)\boldsymbol{\Sigma}_k \boldsymbol{J}_f(\hat{s}_k, a_k)^\top$    $\triangleright$ node attribute of $\hat{s}_{k+1}$, Equation 6
12:     $\mathbb{V}[R_k] \leftarrow \sigma_R^2(\hat{s}_k, a_k) + \boldsymbol{J}_r(\hat{s}_k, a_k)\boldsymbol{\Sigma}_k \boldsymbol{J}_r(\hat{s}_k, a_k)^\top$    $\triangleright$ node attribute of $\hat{s}_{k+1}$, Equation 7
13:     $\mathbb{V}[V_{k+1}] \leftarrow \sigma_V^2(\hat{s}_{k+1}) + \boldsymbol{J}_v(\hat{s}_{k+1})\boldsymbol{\Sigma}_{k+1} \boldsymbol{J}_v(\hat{s}_{k+1})^\top$      $\triangleright$ Equation 8
14:     BACKUP($\hat{s}_{k+1}, \mathbb{E}[V_{k+1}], \mathbb{V}[V_{k+1}]$)      $\triangleright$ updates the tree values & variances

15: **function** BACKUP(node $\hat{s}_{k+1}$, return-mean $\mathbb{E}[\mathcal{V}_{k+1}]$, return-uncertainty $\mathbb{V}[\mathcal{V}_{k+1}]$)
16:     $\hat{s}_k, a_k, \mathbb{E}[\mathcal{V}_k] \leftarrow$ Execute unmodified MCTS backup step (updates $q(\hat{s}_k, a_k)$ and $N(\hat{s}_k, a_k)$)
17:     $\mathbb{V}[\mathcal{V}_k] \leftarrow \mathbb{V}[R_k] + \gamma^2 \mathbb{V}[\mathcal{V}_{k+1}]$      $\triangleright$ uses node-attribute $\mathbb{V}[R_k]$, Equation 10
18:     $\sigma_q^2(\hat{s}_k, a_k) \leftarrow \sigma_q^2(\hat{s}_k, a_k) + \frac{\mathbb{V}[\mathcal{V}_k] - \sigma_q^2(\hat{s}_k, a_k)}{N(\hat{s}_k, a_k)}$      $\triangleright$ node attribute of $\hat{s}_{k+1}$, Equation 12
19:     **if** $k > 0$ **then** BACKUP($\hat{s}_k, \mathbb{E}[\mathcal{V}_k], \mathbb{V}[\mathcal{V}_k]$)      $\triangleright$ updates the tree values & variances

---

### 3.3 ESTIMATING EPISTEMIC UNCERTAINTY IN PLANNING

Epistemic uncertainty estimation techniques in RL are designed to evaluate uncertainty on predictions in the true observation space of the environment (Osband et al., 2018; Burda et al., 2019). These methods translate naturally into planning with transition models that operate in the environment's observation space, such as AlphaZero where the dynamics are given, or when a learned transition model predicts environmental observations. However, when the latent state space $\hat{\mathcal{S}}$ is not identical to the observation space, novelty estimated in latent space may not reflect the novelty in the true state space. Specifically, before the first observation of a non-zero reward, *value-equivalent* models (such as used by MuZero) may abstract all states in sparse-reward environments into one constant representation that supports the value prediction of zero. As a result, all states (even unobserved states) may be associated with the same novelty of zero in the latent space. This problem can be circumvented by driving reconstruction losses (see Section 2.2) through the transition model, incentivizing the learned model to distinguish between unique states, or by learning an auxiliary dynamics model which does not need to be robust but only distinguish between novel and observed starting-states and action sequences. Variations of these methods have been used successfully by Henaff (2019) and Sekar et al. (2020).

To estimate the novelty of states in the true state space of the environment (whether the model is learned or provided) we chose the lightweight novelty estimator RND (see Section 2.3 and Appendix D.4 for additional details) for its expected reliability in detecting unobserved states. To evaluate E-MCTS with the value-equivalent dynamics model of MuZero we provide the agent with reliable (but unrealistic) transition uncertainty in the form of state-action visitation counts in the true state space $\mathcal{S}$ (see Appendix D.5 for additional details). To estimate the value uncertainty at the leaf $\sigma_V^2(\hat{s}_T)$ we use a UBE network-head (see Section 2.3) for all three transition models (given, learned in the true state space, value-equivalent learned in latent space). We allow the gradients with respect to the UBE head to pass through and train the value-equivalent learned transition model, similarly to the gradients of the value, policy and reward functions (see Appendix D.2 for additional details).

## 4 RELATED WORK

Different faces of the idea of leveraging planning with learned dynamics models for exploration have been investigated by a range of previous works, such as Yi et al. (2011), Hester & Stone (2012), Shyam et al. (2019), Sekar et al. (2020), Lambert et al. (2022) and Henaff (2019). Among a range of differences, these methods are not tailored for MCTS or deterministic dynamics's models MCTS algorithms, which are a very strong class of MBRL algorithms. We add to this line of work E-MCTS: tailored for MCTS (and planning trees in general), lightweight and applicable to deterministic models by approximating and propagating the variance directly resulting only in a constant increase in computation cost to MCTS. Moerland et al. (2020) identify that the further a state is from a terminal state in the MCTS planning tree, the more uncertainty should be associated with it in planning, and utilizes this uncertainty to bias search in MCTS. POMCP (Silver & Veness, 2010), POMCPOW (Sunberg & Kochenderfer, 2018) and BOMCP (Mern et al., 2021) extend MCTS to POMDPs with a probabilisticly modelled Bayesian belief state at the nodes using a probabilistic model, while Stochastic MuZero Antonoglou et al. (2021) extended MuZero to the stochastic setting by replacing $f$ with a Vector Quantised Variational AutoEncoder (van den Oord et al., 2017). Epistemic uncertainty is not distinguished explicitly or used for exploration. A common uncertainty / novelty estimation alternative to RND Burda et al. (2019) are ensembles Lakshminarayanan et al. (2016); Ramesh et al. (2022). The uncertainty measure is usually the disagreement between the ensemble's predictions. Bootstrapped DQN (BDQN, Osband et al., 2016; 2018) is an effective model-free deep exploration approach that relies on the epistemic uncertainty estimated by an ensemble to drive exploration. Wasserstein Temporal Difference (WTD, Metelli et al., 2019) offers an alternative to UBE O'Donoghue et al. (2018) for propagating epistemic uncertainty in TD-learning, using Wasserstein Barycenters Agueh & Carlier (2011) to update a posterior over $Q$ functions in place of a standard Bayesian update. UBE was criticized by Janz et al. (2019) for having unnecessary properties as well as being insufficient for deep exploration with posterior-sampling based RL (PSRL, Osband et al., 2013). These shortcomings however do not influence UCB-based exploration algorithms which E-MCTS can be classified as. Pairing with UBE thus enables E-MCTS to benefit from the strengths of UBE (such as uncertainty propagation, as discussed by Janz et al., 2019) while avoiding the shortcomings identified in the paper.

## 5 EXPERIMENTS

We evaluate the following hypotheses: **H1** E-MCTS successfully propagates epistemic uncertainty in planning. **H2** Planning in MCTS with an optimistic objective (Equation 11) is able to achieve deep exploration. **H3** Planning can be leveraged for uncertainty estimation that improves over non-planning-based uncertainty estimation, even with learned dynamics models. We use BSUITE's (Osband et al., 2020) hard exploration benchmark Deep Sea. The Deep Sea environment encapsulates some of the hardest challenges associated with exploration: The probability of finding the unique optimal action trajectory through random action selection decays exponentially with the size of the environment. Every transition in the direction of the goal receives a negative reward that is negligible in comparison to the goal reward, but is otherwise the only reward the agent sees discouraging exploration in the direction that leads to the goal. Finally, the action mappings are randomized such that the effect of the same action is not the same in every state, preventing the agent from generalizing across actions. Three variations of the transition model $f$ are investigated: (i) An AlphaZero model. (ii) A MuZero model. (iii) An anchored model (dynamics trained exclusively with a reconstruction loss). The reward $r$, value $v$ and policy $\pi$ functions are always trained in the MuZero manner, using the framework of EfficientZero (Ye et al., 2021). For implementation details see appendices D.3 D.2 and D.5. We compare four exploration methods: (i) **E-MCTS** (ours). (ii) An Alpha/MuZero agent that uses **UBE** predictions post-planning (see Appendix D.8 for details). (iii) The Alpha/MuZero exploration baseline which is **uninformed** with respect to epistemic uncertainty. (iv) Model-free Bootstrapped DQN (**BDQN**, Osband et al., 2016). The results are presented in Figures 1 and 2. E-MCTS demonstrates reliable uncertainty propagation through successful deep exploration with all three transition models, supporting hypotheses H1 & H2, as well as outperforms the UBE baseline in all three models, demonstrating improvement from planning with propagated uncertainty, supporting hypotheses H3 (Figure 1). E-MCTS scales very well, sub-exponentially as expected (Figure 2, left). Since exploitation and exploration episodes alternate, the exploration parameter $\beta$ need only be large enough to induce sufficient exploration to solve Deep Sea, resulting in low average regret across a wide range of values of $\beta$ (Figure 2, right). Figure 3 demonstrates the reliability of the uncertainty estimated by E-MCTS by comparing it with inverse-counts as ground-truth. As expected, the uncertainty diminishes monotonically throughout training for all visited states.

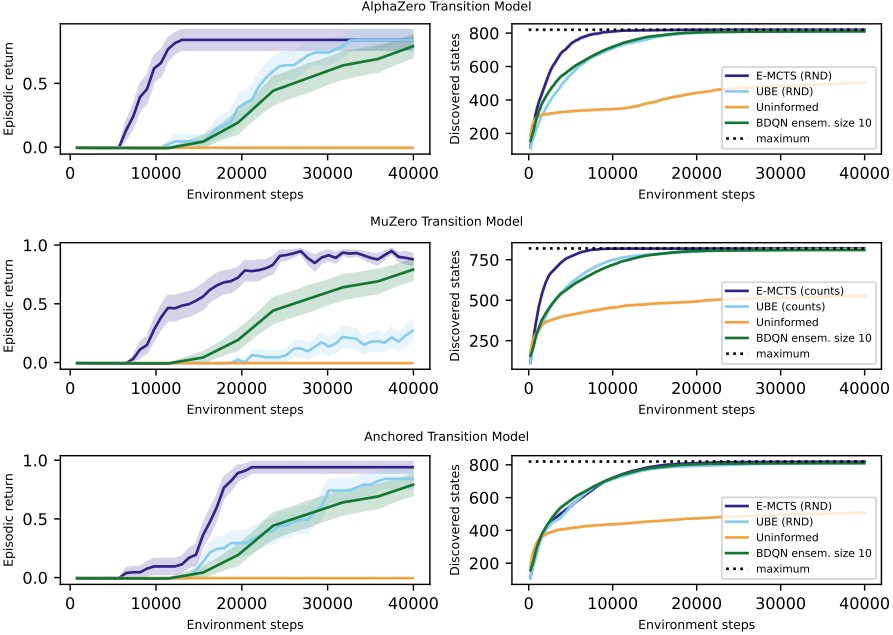

Figure 1: Deep Sea 40 by 40, mean and standard error for 20 seeds. Rows: Different transition models. Left: episodic return in evaluation vs. environment steps. Right: exploration rate (number of discovered states vs. environment steps).

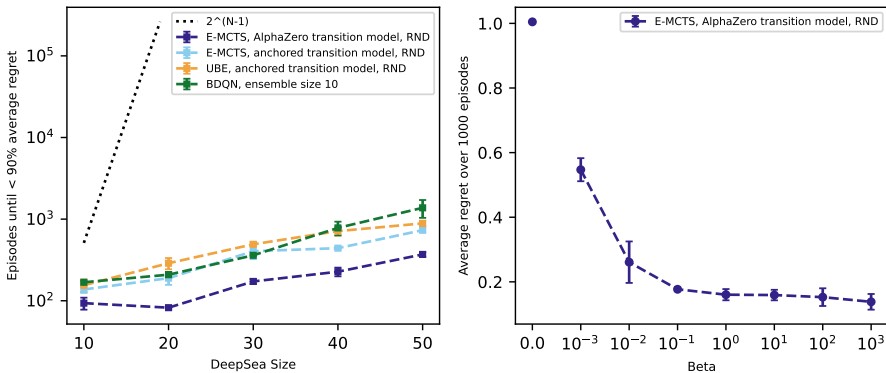

Figure 2: Left: Scaling to growing Deep Sea sizes. Mean of 5 seeds with standard error. Right: The effect of the exploration perparameter $\beta$ for Deep Sea 30 by 30. Mean of 3 seeds with standard error.

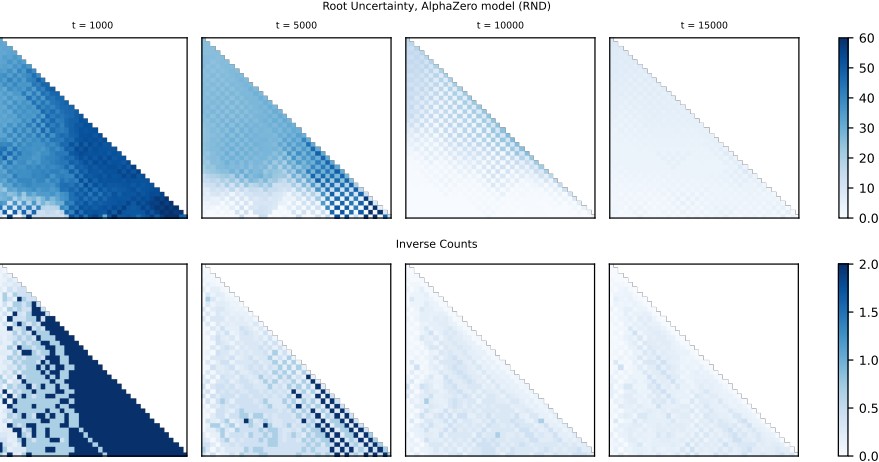

Figure 3: Heat maps over states in DeepSea 40 by 40 (lower triangle) at different times (columns) during an example training run of E-MCTS with an AlphaZero transition model. Upper row: value uncertainty at the E-MCTS root node. Lower row: inverse visitation counts as reliable local uncertainty. Score of 2.0 represents unvisited.

## 6 CONCLUSIONS AND FUTURE WORK

In this work we present E-MCTS, a novel method for incorporating epistemic uncertainty into MCTS. We use E-MCTS to modify the planning objective of MCTS to an exploratory objective to achieve deep exploration with MCTS-based MBRL agents. We evaluate E-MCTS on the Deep Sea benchmark, which is designed to be a hard exploration challenge, where our method yields significant improvements in state space exploration and uncertainty estimation. In addition, E-MCTS demonstrates the benefits of planning for exploration by empirically outperforming non-planning deep exploration baselines. The framework of E-MCTS provides a backbone for propagating uncertainty in other tree-based planning methods, as well as for the development of additional approaches to harnessing epistemic uncertainty. For example: (i) With E-MCTS, it is possible to plan with a conservative objective by discouraging uncertain decisions to improve reliability in the face of the unknown, which is paramount in the offline-RL setting. (ii) E-MCTS can be used to avoid planning into trajectories that increase epistemic uncertainty in value prediction, with the aim of achieving more reliable planning. (iii) Down-scaling of epistemically-uncertain targets has been used by Lee et al. (2021) and Wu et al. (2021) to improve the learning process of online and offline RL agents respectively. Given the advantages in exploration, it stands to reason that the improved value-uncertainty estimates from E-MCTS can benefit those approaches as well.

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

## A  LAW OF TOTAL VARIANCE

The *law of total variance* for two continuous random variables $X$ and $Y$ can be derived as follows:

$$
\begin{aligned}
\mathbb{V}_Y[Y] \;=\;& \int (Y - \mathbb{E}_Y[Y])^2 \, p(Y) \, dY \;=\; \iint (Y - \mathbb{E}_Y[Y])^2 \, p(X,Y) \, dX \, dY \\
=\;& \iint (Y - \mathbb{E}_Y[Y])^2 \, p(Y|X) \, p(X) \, dX \, dY \;=\; \mathbb{E}_X\Big[\mathbb{E}_Y\big[(Y - \mathbb{E}_Y[Y])^2 \big| X\big]\Big] \\
=\;& \mathbb{E}_X\Big[\mathbb{E}_Y\big[(Y - \mathbb{E}_Y[Y|X] + \mathbb{E}_Y[Y|X] - \mathbb{E}_Y[Y])^2 \big| X\big]\Big] \\
=\;& \mathbb{E}_X\Big[\mathbb{E}_Y\big[\underbrace{(Y - \mathbb{E}_Y[Y|X])^2 \big| X}_{\mathbb{V}_Y[Y|X]}\big]\Big] + 2\,\mathbb{E}_X\Big[\big(\underbrace{\mathbb{E}_Y[Y|X] - \mathbb{E}_Y[Y|X]}_{0}\big)\big(\mathbb{E}_Y[Y|X] - \mathbb{E}_Y[Y]\big)\Big] \\
& + \underbrace{\mathbb{E}_X\Big[\big(\mathbb{E}_Y[Y|X] - \mathbb{E}_Y[Y]\big)^2\Big]}_{\mathbb{V}_X[\mathbb{E}_Y[Y|X]]} \;=\; \mathbb{V}_X\big[\mathbb{E}_Y[Y|X]\big] + \mathbb{E}_X\big[\mathbb{V}_Y[Y|X]\big]
\end{aligned}
$$

## B  FIRST ORDER TAYLOR APPROXIMATION OF VARIANCE OF NEXT STATE

This appendix derives Equation 6:

$$
\begin{aligned}
\boldsymbol{\Sigma}_{k+1} \;:=\; \mathbb{V}[\hat{S}_{k+1}] \;=\;& \underbrace{\mathbb{E}_{\hat{S}_k}\big[\mathbb{V}_{\hat{S}_{k+1}}[\hat{S}_{k+1}|\hat{S}_k, a_k]\big]}_{\boldsymbol{\Sigma}(\hat{s}_k, a_k)} \;+\; \underbrace{\mathbb{V}_{\hat{S}_k}\big[\mathbb{E}_{\hat{S}_{k+1}}[\hat{S}_{k+1}|\hat{S}_k, a_k]\big]}_{\boldsymbol{J}_f(\hat{s}_k, a_k)\,\boldsymbol{\Sigma}_k\,\boldsymbol{J}_f(\hat{s}_k, a_k)^\top} \\
\approx\;& \boldsymbol{\Sigma}(\hat{s}_k, a_k) \;+\; \boldsymbol{J}_f(\hat{s}_k, a_k)\,\boldsymbol{\Sigma}_k\,\boldsymbol{J}_f(\hat{s}_k, a_k)^\top\,.
\end{aligned}
$$

First note that $\mathbb{E}[\hat{S}_k] = \hat{s}_k$, $\mathbb{E}[\hat{S}_{k+1}|\hat{S}_k, a_k] = f(\hat{S}_k, a_k)$ and $\mathbb{V}[\hat{S}_{k+1}|\hat{S}_k, a_k] = \boldsymbol{\Sigma}(\hat{S}_k, a_k)$ by definition. The first term is approximated using first order Taylor approximation as follows:

$$
\mathbb{E}[\mathbb{V}[\hat{S}_{k+1}|\hat{S}_k, a_k]] \;=\; \mathbb{E}[\boldsymbol{\Sigma}(\hat{S}_k, a_k)] \approx \mathbb{E}[\boldsymbol{\Sigma}(\hat{s}_k, a_k) + (\hat{S}_k - \hat{s}_k)^\top \boldsymbol{J}_{\boldsymbol{\Sigma}}(\hat{s}_k, a_k)^\top] \;=\; \boldsymbol{\Sigma}(\hat{s}_k, a_k)\,.
$$

The second term is approximated in a similar manner:

$$
\mathbb{V}[\mathbb{E}[\hat{S}_{k+1}|\hat{S}_k, a_k]] = \mathbb{V}[f(\hat{S}_k, a_k)] \approx \mathbb{E}\big[\big(\underbrace{(\hat{S}_k - \hat{s}_k)^\top \boldsymbol{J}_f(\hat{s}_k, a_k)^\top}\big)^2\big] = \boldsymbol{J}_f(\hat{s}_k, a_k)\boldsymbol{\Sigma}_k\boldsymbol{J}_f(\hat{s}_k, a_k)^\top,
$$

due to

$$
f(\hat{S}_k, a_k) - \mathbb{E}[f(\hat{S}_k, a_k)] \;\approx\; (\hat{S}_k - \hat{s}_k)^\top \boldsymbol{J}_f(\hat{s}_k, a_k)^\top - \underbrace{\mathbb{E}[(\hat{S}_k - \hat{s}_k)^\top]}_{0}\, \boldsymbol{J}_f(\hat{s}_k, a_k)^\top
$$

## C  ADDITIONAL RESULTS

We include a table evaluating interactions-to-goal on deep-sea 40 by 40 for all investigated transition models and the three MCTS agents: E-MCTS (ours), UBE and the uninformed baseline. The results demonstrate that even when the learned dynamics model is not designed for planning (anchored model, third block, Table 1), E-MCTS is able to find the goal much faster. Not due to faster-exploration per-se (1, third row, the exploration rate is very similar across baselines), but due to apparent higher reliability of the uncertainty prediction, resulting in *reliable* exploration throughout the entire state-action space.

Table 1: Number of environment steps until the first visitation to the goal transition.

|  | Exploration | Average steps to goal transition for seeds that discovered goal $\pm$ STD | % seeds that discovered goal |
|---|---|---|---|
| AlphaZero Model + RND | E-MCTS | **10539** $\pm$ 9006 | 94% of 35 seeds |
|  | UBE | 22801 $\pm$ 7514 | 91% of 35 seeds |
|  | Uninformed | - | 0% of 20 seeds |
| MuZero Model + Counts | E-MCTS | **14339** $\pm$ 6845 | 100% of 23 seeds |
|  | UBE | 29945 $\pm$ 8113 | 57% of 21 seeds |
|  | Uninformed | - | 0% of 20 seeds |
| Anchored Model + RND | E-MCTS | **15241** $\pm$ 3236 | 95% of 20 seeds |
|  | UBE | 22497 $\pm$ 6645 | 85% of 20 seeds |
|  | Uninformed | - | 0% of 20 seeds |

# D IMPLEMENTATION DETAILS

## D.1 TARGETS

In MuZero, the value targets $v_{t+k}^{\text{MCTS}}$ for the prediction of value of latent state $\hat{s}_t^k$ that matches true state $s_{t+k}$ are computed as an $n$-step TD target:

$$v_{t+k}^{\text{MCTS}} = \sum_{i=0}^{n-1} \gamma^i r_{t+k+i} + \gamma^n v_{t+k+n}^{\text{MCTS}}$$

Where $v_{t+k+n}^{\text{MCTS}}$ can be computed in one of two ways:

(i) The value of the root of an MCTS tree computed for state $s_{t+k+n}$.

(ii) A prediction of the value network $v$ for latent state $\hat{s}_{t+k+n}^0$.

Method (i) is expected to result in better value targets, but is more expensive computationally. Method (ii) is significantly cheaper computationally, but might hinder learning through the lack of value improvement (a max operator) on the value bootstrap. We refer to (i) as *root-based targets*.

The UBE target $u_{t+k}^{\text{target}}$ for the prediction of value-uncertainty from the UBE head $u(\hat{s}_t^k)$ is computed a similar manner:

$$u_{t+k}^{\text{target}} = \sum_{i=0}^{n-1} \gamma^{2i} \sigma^2(\hat{s}_{t+k+i}^0, a_{t+k+i}) + \gamma^{2n} u_{t+k+n}$$

Analogous to the value target, the bootstrap $u_{t+k+n}$ can be computed in two different ways:

(i) When E-MCTS is used, the target can be computed similarly to the MuZero value target, as the epistemic uncertainty of the root of an E-MCTS tree computed for state $s_{t+k+n}$. This tree can plan for an exploitatory objective (equation 1) to estimate the uncertainty of the value $V^\pi(s_{t+k+n})$, an exploratory objective (equation 11) to estimate the uncertainty of the value associated with the exploration policy, or even an uncertainty-maximizing objective:

$$a_k := \underset{a_k}{\arg\max} \sqrt{\sigma_q^2(\hat{s}_k, a_k)} + C\sqrt{\frac{2\log(\sum_{a'} N(\hat{s}_k, a'))}{N(\hat{s}_k, a_k)}}$$

Where the $q$ term has been dropped entirely as an optimistic bound over the uncertainty to encourage exploration. Similarly, we refer to using as target the E-MCTS uncertainty prediction at the root as a root-based target. In our experiments, when UBE root-based target were used, we have used the uncertainty-maximizing objective.

(ii) When E-MCTS is not used, the UBE bootstrap $u_{t+k+n}$ is computed as the maximum UBE over possible actions from state $s_{t+k+n}$:

$$u_{t+k+n} = \underset{a_{t+k+n}}{\max} \sigma^2(\hat{s}_{t+k+n}^0, a_{t+k+n}) + \gamma^2 u(f(\hat{s}_{t+k+n}^0, a_{t+k+n}))$$

These targets were used for all UBE-only agents, and for the E-MCTS agents that did not use root-based targets.

In all experiments we have used $n = 1$ (one-step targets) for the UBE targets.

In MuZero, the reward and value predictions $r(\hat{s}_t^k, a_{t+k}), v(\hat{s}_t^k)$ are represented as a discrete probability distribution over a range of discrete values $[-M, M], M \in \mathbb{N}$. To transform the scalar value and reward targets to a categorical representation of the same representation format, a transformation function $\phi(x)$ is used, transforming a real number $x$ into a categorical representation through a linear interpolation between its adjacent integers.

## D.2 Losses

The original MuZero algorithm uses three loss functions:

$$\mathcal{L}_r := \frac{1}{|\mathcal{B}|} \sum_{t \in \mathcal{B}} \sum_{k=0}^{l-1} \phi(r_{t+k})^\top \log r(\hat{s}_t^k, a_{t+k})$$

$$\mathcal{L}_v := \frac{1}{|\mathcal{B}|} \sum_{t \in \mathcal{B}} \sum_{k=0}^{l-1} \phi(v_{t+k}^{\text{MCTS}})^\top \log v(\hat{s}_t^k)$$

$$\mathcal{L}_\pi := \frac{1}{|\mathcal{B}|} \sum_{t \in \mathcal{B}} \sum_{k=0}^{l-1} \pi^{\text{MCTS}}(s_{t+k})^\top \log \pi(\hat{s}_t^k)$$

Where $\mathcal{B} \equiv \{s_t, a_t, r_t, s_{t+1}, a_{t+1}, \ldots, s_{t+l}\}_{t \in \mathcal{B}}$ is a training batch containing $b$ trajectories of length $l$ sampled from different episodes, $r_{t+k}$ is the true reward observed in the environment, $r(\hat{s}_t^k, a_k), v(\hat{s}_t^k), \pi(\hat{s}_t^k)$ are respectively the reward value and policy predictions for latent state $\hat{s}_t^k$ (and action $a_{t+k}$ when appropriate). $\pi^{\text{MCTS}}(s_{t+k})$ is a discrete probability distribution computed based on the normalized visitation counts to the children of an MCTS root computed at state $s_{t+k}$ (see Equation 3).

In MuZero the gradient from the losses $\mathcal{L}_r, \mathcal{L}_v, \mathcal{L}_\pi$ propagates through the transition model $f$ and are the only learning signal that is used to train the model. For the anchored model (see Section 5) we use an additional reconstruction loss:

$$\mathcal{L}_{re} := \frac{1}{|\mathcal{B}|} \sum_{t \in \mathcal{B}} \sum_{k=0}^{l-1} ||\hat{s}_t^k - s_{t+k}||^2$$

Which can alternatively be thought of as a consistency loss, where $g$ is the identity function. The mean squared error loss is denoted with $\mathcal{L}_{\text{MSE}}$. To estimate value-uncertainty at the leaves, we train a UBE function $u$ with a UBE loss $\mathcal{L}_u$:

$$\mathcal{L}_u := \frac{1}{|\mathcal{B}|} \sum_{t \in \mathcal{B}} \sum_{k=0}^{l-1} \phi(u_{t+k}^{\text{target}})^T \log \hat{u}_t^k$$

The final loss is computed as:

$$\mathcal{L} := \lambda_r \mathcal{L}_r + \lambda_v \mathcal{L}_v + \lambda_\pi \mathcal{L}_\pi + \lambda_u \mathcal{L}_u$$

Where the coefficients $\lambda_r, \lambda_v, \lambda_\pi, \lambda_u$ are used to weigh the relative effects the individual components of the loss have on the learned transition model $f$. When $\mathcal{L}_{re}$ was used (the anchored model in Section 5), the model parameters of $f$ were affected only by $L_{re}$, through a second backwards pass.

## D.3 Different Dynamics Models

We describe the three transition models used in 5 in more detail. The AlphaZero dynamics model is a true model of the dynamics of the environment, in the true state space of the environment. When planning with this model local uncertainty is estimated with RND and value-uncertainty is estimated with UBE. The MuZero model is a value-equivalent model in latent space. $g, f$ are learned by the agent during training from the value, policy, reward and UBE losses. When planning with this model local uncertainty is estimated with state-visitation-counts (see D.5 and value-uncertainty is estimated with UBE. The anchored-MuZero transition model trained only to predict the true transition dynamics of the environment through a reconstruction loss $L_{re}^k$ (see Appendix D.2). When planning with this model local uncertainty is estimated with RND and value-uncertainty is estimated with UBE.

## D.4 Planning with Random Network Distillation Based Epistemic Uncertainty

Many popular novelty-estimators in deep RL (such as RND, or even deep-ensembles Osband et al., 2018) do not directly provide a reliable variance estimate. Many deep ensemble methods for example rely on ensemble disagreement (Sekar et al., 2020), but do not assume that the variance in the ensemble approximates the variance of a distribution, but rather should be high on unknown and low on known inputs. This problem is exacerbated when a covariance *matrix* needs to be approximated (for example, for Equation 6). To circumvent this limitation of standard and popular uncertainty estimation methods we estimate the uncertainty in latent state $\Sigma_k$ and the uncertainty in the predictions based on latent state $\sigma_R$ together into one score. More specifically, we use $\Sigma_k = 0$

and $\sigma_V(\hat{s}_k) = \max\left(L_{rnd}(\hat{s}_{k-1}, a_{k-1}), u(f(\hat{s}_{k-1}, a_{k-1}))\right)$. This choice is sufficient for E-MCTS to significantly improve over a comparable non-planning deep exploration baselines, see Section 5.

When the planning is done with a true model, the agent has access to the true states $s_{t+k}$ and can use RND to evaluate transition uncertainty over the state action pair $(s_{t+k}, a_{t+k})$ directly. When the planning is done with the anchored model, the latent states outputted by the transition model $\hat{s}_t^k$ approximate the true states $s_{t+k}$ which allows us to use RND over $(\hat{s}_t^k, a_{t+k})$. In both cases, RND is trained only over the observed transitions $(s_{t+k}, a_{t+k})$, not latent state representations $(\hat{s}_t^k, a_{t+k})$, to achieve the objective of yielding large RND prediction errors the further the latent state prediction $\hat{s}_t^k$ is from observed state $s_{t+k}$.

## D.5 PLANNING WITH VISITATION-COUNTS BASED EPISTEMIC UNCERTAINTY

When planning with the abstracted model, we provide the agent with access to two additional mechanisms that are used only for local uncertainty estimation: the true model $F(s_t, a_t)$ of the environment and a state-action visitation counter $C(s_t, a_t)$. During planning, the true transition model follows the planning decisions $a_{t:t+k}$ and keeps track of the true state $s_{t+k}$. When the agent evaluates the local uncertainty with transition $(\hat{s}_t^k, a_{t+k})$ the true model provides the matching true state $s_{t+k}$ to the visitation counter, which produces the local uncertainty based on the following formula:

$$\sigma^2(s_{t+k}, a_{t+k}) = \frac{1}{C(s_{t+k}, a_{t+k}) + \epsilon}$$

Where $0 < \epsilon \leq 1$ is a constant and $C(s_{t+k}, a_{t+k})$ counts the number of times the state action pair $(s_{t+k}, a_{t+k})$ has been observed in the environment. This allows us to evaluate the abstracted-model agent in the presence of a reliable source of local uncertainty. The leaf-value uncertainty $u(\hat{s}_t^k)$ (which is the dominating factor in visited areas of the state space, as $\sigma^2(s_{t+k}, a_{t+k}) \to 0$ quickly in observed transitions) relies entirely on the learned UBE function $u$ which operates directly on latent states $\hat{s}_t^k$.

## D.6 SEPARATING EXPLORATION FROM EXPLOITATION

Acting in the environment with a dedicated exploration policy can be expected to result in samples that are very off-exploitation-policy. Learning from very off-policy data is known for causing instability in training even in off-policy agents. To mitigate that, the E-MCTS and only-UBE agents (see section 5) alternate between two types of training episodes: *exploratory episodes* that follow an exploration policy throughout the episode (such as a policy generated by E-MCTS with an exploratory planning objective), and *exploitatory episodes* that follow the standard MuZero exploitation policy throughout the episode. This enables us to provide the agent with quality exploitation targets to evaluate and train the value and policy functions reliably, while also providing a large amount of exploratory samples that explore the environment much more effectively and are more likely to efficiently search for high-reward interactions.

In practice, rather than alternate between exploration and exploitation episodes we run a certain number of episodes in parallel, a certain portion of which are exploitatory and the rest are exploratory. In our experiments the ratio was $50/50$. During exploration episodes, we do not wish to bias the search in the tree with respect to previously tried actions, but rather only with respect to the combination of value and uncertainty (equation 11). We set the policy prediction $\pi(\hat{s}_t^k)$ (see Equation 2) to uniform over all actions, for all $\hat{s}_t^k$ during exploration episodes. In addition, Dirichlet noise was not used to drive exploration in MCTS with the UBE and E-MCTS agents.

## D.7 ENVIRONMENT ADAPTATION

To maintain the exploration difficulty of Deep Sea while reducing numerical challenges, we amplify the goal reward from 1 to 10. To limit the challenge of learning a model that can distinguish between approximately $N^2$ unique states when learning the true dynamics of the environment, while retaining the exploration challenge of searching for one trajectory in a total of $2^N$ trajectories, we choose environment size $N = 40$, for a $(40, 40)$ grid. To further simplify model learning with the anchored model, the representation function $g$ that was used for the anchored model transforms the observations

from 2 dimensional $(N, N)$ one-hot representations to 1 dimensional $(2N)$ representations where the first $N$ entries are a 1-hot vector representing the row and following $N$ entries are a 1-hot vector representing the column. From this perspective, we can view the $\mathcal{L}_{re}$ loss that was used to train the anchored model as a consistency loss between the representation and the state prediction rather than a reconstruction loss. The loss itself is the same loss specified in Appendix D.2.

### D.8 UBE BASELINE

The UBE baseline agent uses MCTS to evaluate the value of actions using MCTS in the same manner as Alpha/MuZero, and explores by taking the action $a_t$ that maximizes the combination of the Q-values approximated by MCTS $q$, local uncertainty $\sigma^2$ and UBE $u$:

$$a_t = \arg\max_a q(\hat{s}_0, a_t) + \beta \sqrt{\sigma^2(\hat{s}_0, a_t) + \gamma^2 u(f(\hat{s}_0, a_t))}. \tag{14}$$

### D.9 COMPUTE

The experiments were run on the [anonymized for review] computation clusters, using any of the following GPU architectures: NVIDIA Quadro K2200, Tesla P100, GeForce GTX 1080 Ti, GeForce RTX 2080 Ti, Tesla V100S and Nvidia A-40. Each seed was ran on one GPU, and was given access to 100 GB of RAM and 16 CPU cores. Total training time was in the range of 12 to 65 hours per seed, depending on GPU architecture and whether root-based targets (see Appendix D.1) which significantly increased training time were used or not.

## E NETWORK ARCHITECTURE & HYPERPARAMETERS

### E.1 HYPERPARAMETER SEARCH

Due to the large number of hyperparameters in the MuZero framework, our optimization process consisted of manual modifications to the hyperparameters used by Ye et al. (2021) with the objective of achieving learning stability on the target environment with the simplest network architectures possible. Two exceptions to this statement are the RND network architecture and scale, and the exploration parameter $\beta$.

The RND architecture was designed with the objective of reliably achieving small RND predictions over observed state-action pairs and large predictions over unobserved state-action pairs. The RND scale was tuned with the objective of achieving local uncertainty measures for unobserved state-action pairs that are significantly larger than the minimum reward of Deep Sea.

The $\beta$ parameter was tuned with the objective that the E-MCTS and only-UBE agents will prioritize exploration of the environment over exploitation until the entire environment has been searched, and was tuned separately for every model.

### E.2 NETWORK ARCHITECTURE

The functions $f, g, r, v, u, \pi, \psi, \psi'$ used fully connected DNNs of varying sizes. The sizes of the hidden layers and output layers are specified in Table 2.

### E.3 HYPERPARAMETER CONFIGURATION

We detail the full set of hyperparameters in Tables 3 and 4. For the BDQN baseline, we used the default implementation in `https://github.com/deepmind/bsuite`, with ensemble size of 10 and matching batch size to E-MCTS: number of unroll steps times batch size $5 \cdot 256 = 1230$. Otherwise, the default hyper parameters were used.

Table 2: Network architecture hyperparameters

| True Model | | |
|---|---|---|
| Function | Hidden Layers Sizes | Output Layer Size |
| f | - | - |
| g | - | - |
| r | [256, 256] | 21 |
| v | [256, 256] | 21 |
| u | [256, 256] | 21 |
| $\pi$ | [256, 256] | 2 |
| Anchored Model | | |
| Function | Hidden Layers Sizes | Output Layer Size |
| f | [1024, 1024, 1024] | 80 |
| g | - | - |
| r | [256, 256] | 21 |
| v | [256, 256] | 21 |
| u | [256, 256] | 21 |
| $\pi$ | [256, 256] | 2 |
| Abstracted Model | | |
| Function | Hidden Layers Sizes | Output Layer Size |
| f | [1024, 1024, 1024] | 100 |
| g | [512, 512] | 100 |
| r | [128, 128] | 21 |
| v | [128, 128] | 21 |
| u | [128, 128, 128] | 21 |
| $\pi$ | [128, 128] | 2 |
| RND network architecture | | |
| Function | Hidden Layers Sizes | Output Layer Size |
| $\psi$ | [1024, 1024] | 512 |
| $\psi'$ | [512] | 512 |

Table 3: Shared across all models and agents

| Parameter | Setting | Comment |
|---|---|---|
| Stacked Observations | 1 | |
| $\gamma$ | 0.995 | |
| Number of simulations in MCTS | 50 | |
| Dirichlet noise ratio ($\xi$) | 0.3 | |
| Root exploration fraction | 0 | |
| Batch size | 256 | |
| Learning rate | 0.0005 | |
| Optimizer | Adam (Kingma & Ba, 2015) | |
| Unroll steps $l$ | 5 | |
| Value target TD steps ($n_v$) | 5 | |
| UBE target TD steps ($n_u$) | 1 | |
| value support size | 21 | |
| UBE support size | 21 | |
| Reward support size | 21 | |
| Reanalyzed policy ratio | 0.99 | See (Ye et al., 2021) |
| Prioritized sampling from the replay | True | See (Schrittwieser et al., 2020) Appendix G |
| Priority exponent ($\alpha$) | 0.6 | See (Schrittwieser et al., 2020) Appendix G |
| Priority correction ($\beta_p$) | $0.4 \rightarrow 1$ | See (Schrittwieser et al., 2020) Appendix G |
| Evaluation episodes | 8 | |
| Min replay size for sampling | 300 | |
| Self-play network updating inerval | 5 | |
| Target network updating interval | 10 | |

Table 4: Specific for models and agents

| Parameter | Setting | | | | | | | | |
| --- | --- | --- | --- | --- | --- | --- | --- | --- | --- |
| | True Model | | | Abstracted Model | | | Anchored Model | | |
| | E-MCTS | UBE | Uninf. | E-MCTS | UBE | Uninf. | E-MCTS | UBE | Uninf. |
| Training steps / environment interactions | 45K | 45K | 45K | 35K | 35K | 35K | 45K | 45K | 45K |
| Reward loss weight $\lambda_r$ | 1 | 1 | 1 | 1 | 1 | 1 | 1 | 1 | 1 |
| Value-loss weight $\lambda_v$ | 0.5 | 0.5 | 0.5 | 0.5 | 0.5 | 0.5 | 0.5 | 0.5 | 0.5 |
| Policy-loss weight $\lambda_\pi$ | 0.5 | 0.5 | 0.5 | 0.5 | 0.5 | 0.5 | 0.5 | 0.5 | 0.5 |
| UBE-loss weight $\lambda_u$ | 0.125 | 0.125 | - | 0.25 | 0.25 | - | 0.125 | 0.125 | - |
| RND scale | 1.0 | 1.0 | - | - | - | - | 0.001 | 0.001 | - |
| Root based targets | False | False | False | True | True | True | False | False | False |
| Disabled policy in exploration | True | True | False | True | True | False | True | True | False |
| Number of parallel episodes | 2 | 2 | 2 | 2 | 2 | 2 | 2 | 2 | 2 |
| Out of are exploration episodes | 1 | 1 | - | 1 | 1 | - | 1 | 1 | - |
| Exploration coefficient $\beta$ | 10 | 10 | - | 1 | 1 | - | 10 | 10 | - |
| Dirichlet noise magnitude $\rho$ | 0 | 0 | 0.25 | 0 | 0 | 0.25 | 0 | 0 | 0.25 |

