# OpenReview forum: "E-MCTS: Deep Exploration by Planning with Epistemic Uncertainty"
_ICLR.cc/2024/Conference — ICLR 2024 Conference Withdrawn Submission_

### Official Review · Reviewer_RXgy · 2023-10-30

**Soundness:** 2 fair
**Presentation:** 1 poor
**Contribution:** 3 good
**Rating:** 3
**Confidence:** 3

**Summary:**

Authors tackle the issue of deep exploration in model-based reinforcement learning agents, such as MuZero and AlphaZero. Deep exploration involves the capacity to evaluate not just the immediate rewards of an action but also its long term consequences. o enable deep exploration within MuZero, the authors introduce the "Epistemic MCTS" algorithm, which leverages epistemic uncertainty to facilitate more intelligent exploration. They test the effectiveness of their approach on deep-sea problem, a simple problem designed to test for deep exploration.

**Strengths:**

While MuZero and AlphaZero excel in various tasks, they often struggle to account for the long-term consequences of their actions, particularly in scenarios with sparse rewards. By enabling deep exploration in MuZero, authors can improve the performance of MuZero on many tasks.

Authors  introduce the Epistemic-MCTS algorithm, which incorporates epistemic uncertainties into the decision-making process when selecting actions. This algorithm holds considerable promise as an independent research contribution.

Authors demonstrate the efficacy of their method on a simple and clear problem, which is greatly appreciated.

**Weaknesses:**

1. Although authors try to make their work more mathematically rigorous, I personally find it too hard to follow. Having a separate section on notation could be very helpful.

2. The concept of reconstruction loss, which is new compared to the MuZero paper, is a noteworthy addition; however, it could benefit from more comprehensive explanation. This issue of less detailed explanations appears in multiple sections of the paper. For instance, in the concluding part of Section 2, the term "local uncertainties" is introduced for the first time, yet it lacks clarity regarding the specific context of "local" and the variables to which the authors are referring to.

3. The proposed algorithm, E-MCTS, necessitates the computation of Jacobians, a process that can be computationally intensive. This computational demand may limit the practical applicability of the algorithm to more complex problems.

4. I find the experimental setup to be somewhat constrained. While the deep-sea problem serves as a suitable testbed for assessing deep exploration, it would have been valuable to investigate whether their algorithm negatively impacts the existing capabilities of the MuZero algorithm.

**Questions:**

1. In Section 2.3 authors mention that they use RND and visit counts to approximate epistemic uncertainty. However the details about how and where these values are computed is not mentioned. It is not clear if the RND loss presented in Section 2.3 is used to estimate uncertainties in rewards or values or policies. Can authors clarify this? Another work on epistemic uncertainty estimation which could be relevant to authors is epistemic neural networks (https://arxiv.org/pdf/2107.08924.pdf)

2. Can authors clarify what "local uncertainties" mean at the end of Section 2.3, above the equation.

3.  U^\pi was introduced at the end of Section 2.3, but I wasn't able to find where it was being used. Can you clarify this?

4. In the second paragraph of Section 3, when talking about how to reduce epistemic uncertainty, authors say "2. planning in directions where the agent is more epistemically-certain". It is not clear which variable's epistemic uncertainty authors are referring to. Intuitively, taking a path with more epistemic uncertainty should reduce epistemic uncertainty. Can authors kindly clarify the statement?

5. The authors use Jacobians at various points within their algorithm, for instance, in Equation 6. Based on my understanding, Jacobian computations and the matrix multiplication, such as in the second term of Equation 6, can be computationally very demanding. It would be insightful to understand the strategies the authors employ to manage and mitigate these computational challenges.

6. In Algorithm 1, authors update terms like Sigma_k, in line 11 of the algorithm. Can authors clarify if the updates being performed locally or if the changes in lines 11-13 and lines 17-18 global?

7. Have the authors compared their algorithm on tasks other than deep sea and have they observed any regression when compared to baseline algorithms which use standard MCTS?

8. It looks like the default parameters of BootDQN were used in the experiments in Section 5. Since, the default parameters in bsuite were probably chosen to perform well on all bsuite tasks, it might be useful to tune hyperparamters such as ensemble size, num of sgd steps after each env steps, learning rate and prior scale.


Some suggestions:
1. it would be helpful to have a notation section, as many variables are based on the same letters (ex: v is used in various forms)
2. It might be helpful to include some diagrams on how different components fit together. This can be done for both training and inference.

---

> ### Author Response · Authors · 2023-11-19
>
> Dear reviewer,
>
> thank you for the detailed review and many great comments, questions and suggestions, and especially for the many on-point clarification questions which point out directly where our explanations are not clear and will help us significantly improve the presentation of the paper.
>
> 1. Thanks for pointing that out, we will make sure that the below is made clear and explicit in the paper.
> RND / counts are used to estimate the epistemic uncertainty in reward function and / or transition function locally for a transition $(s,a)$. As opposed to:
>
> 2. i. the uncertainy for $(S',a)$, where the state itself is uncertain (in Equation 7, for example), and ii. the uncertainty in value $\sigma_V(s_T)$, which must take into account uncertainty from future decisions and not just the novelty of $s_T$ locally. Which is why we refer to counts / RND as local measures of uncertainty, or measures of local uncertainty (local uncertainties).
>
> 3. The UBE prediction is used to approximate the uncertainty of value $ U^\pi(s_T) \approx \sigma^2_V(s_T) $ at the leaves $s_T$. We mention this in section 3.3 in the second paragraph and we will make sure this is made more clear.
>
> 4. We refer to the epistemic uncertainty of backups $Var[\nu^i]$ during planning, where the $Var[\nu^i]$ will grow when the agent plans in untrained areas, and shrink when it plans in trained areas. Does that make sense?
> We will make sure it is made clear in the paper (or removed).
>
> 5. We bundle the epistemic uncertainty for reward and transition together into one score, which relates to the observation in 1., that they are both functions of the novelty of the transition locally.
> We propagate this score as if the transitions are certain, thus dropping the Jacobians' computation in practice.
> While this may seem crude, we believe it is reasonable in the deterministic-transition environments that E-MCTS is tailored for.
> If the epistemic uncertainty estimate is reliable, this allows us to give maximum uncertainty score to all trajectories that lead the agent into at least one unobserved transition, and a reasonable score through all observed transitions.
> We currently touch on this in Appendix D.4, but we will make sure we modify the presentation of the uncertainty propagation in the paper to include this discussion.
>
> 6. All updates in E-MCTS are local per node along the backup path $\nu^i$ of planning step $i$ (similar to MCTS). We will make sure this is clear in the paper and in the pseudocode.
>
> 7. We have not compared to standard MCTS, but only focused on turning the learning-based Alpha/MuZero into an epistemic algorithm, and showing benefits from that in the form of deep exploration.
> Prior to this question, we have not considered as part of the motivation for this work the advantage of learned value functions over rollouts, that rollouts-based MCTS cannot "learn" from one episode to the next, but Alpha/MuZero can, and thus epistemic Alpha/MuZero can explore an environment much more efficiently over time than MCTS. We believe that this is another setting where we can motivate the benefit of epistemic learning-based algorithms over non-epistemic, non-learning based planning algorithms.
> We hope to include results on more complex environments with the next iteration of the paper.
>
> Does this address the reviewer's questions?

---

> ### Comment · Reviewer_RXgy · 2023-11-22
> **Re: authors response**
>
> I thank the authors for their time and effort in addressing the issues I raised. I find the work interesting, and I believe that refining the paper's presentation and incorporating experiments that compare against MuZero on environments which doesn't require deep exploration could significantly enhance its overall quality.

---

### Official Review · Reviewer_9sR2 · 2023-10-30

**Soundness:** 2 fair
**Presentation:** 2 fair
**Contribution:** 3 good
**Rating:** 5
**Confidence:** 5

**Summary:**

This paper presents Epistemic-MCTS (E-MCTS), a method for incorporating epistemic uncertainty into AlphaZero-like model-based RL with MCTS. The goal is to encourage the selection of actions for decisions with more epistemic uncertainty caused by epistemic errors from the learned model. This approach facilitates exploration in states that require more interactions, thereby inducing deep exploration. Comparing to the baselines, E-MCTS outperforms in the investigated exploration benchmark (e.g., Deep Sea).

**Strengths:**

1. The idea of considering epistemic uncertainty in MCTS selection is interesting and reasonable for reinforcement learning (RL).
2. E-MCTS claims to provide a method to further improve the performance of existing model-based RL algorithms with MCTS, such as AlphaZero and MuZero.
3. This paper provides insights into estimating epistemic uncertainty using a recursive approximation of variance.

**Weaknesses:**

# Major
1. Although the paper mentions some literature reviews about uncertainty, it lacks a more comprehensive survey, particularly in the early deep reinforcement learning (DRL) research. Two foundational works in early DRL, VIME (Variational Information Maximizing Exploration) and IDS (Information-Directed Exploration for Deep Reinforcement Learning), should be included to strengthen the survey.
2. In Section 2.2, there is a concern regarding soundness. The original AlphaZero/MuZero models do not include a reconstruction loss. It would be more appropriate to refer to other methods, such as Dreamer or EfficientZero, that address this issue.
3. Regarding the experiments, it is noticeable that Deep Sea is a deterministic environment, whereas there is a stochastic variant available in bsuite. One may wonder why these experiments primarily focus on the deterministic version. This choice is particularly interesting given the presence of various sources of uncertainty in stochastic environments. It raises questions about the suitability of E-MCTS in stochastic environments and whether it can outperform AlphaZero/MuZero in complex scenarios for which the latter were specifically designed. Additionally, it's worth noting that we lack a straightforward MCTS baseline that does not suffer from epistemic uncertainty issues. It is possible that a simple MCTS approach may outperform AlphaZero/MuZero in this specific context, which could weaken the empirical evidence.
# Minor
1. Equation 3: It is unclear why there is a $max_\pi$ before $V^{\pi}(s_t)$ since there is no policy selection process among a set of policies. This inconsistency should be addressed.
2. Equation 11: The later part of the equation refers to $a_k$ without prior definition. It seems that all $a_k$ should be $a$. Additionally, the "$argmax$" function should be enclosed in parentheses to avoid confusion.
3. Regarding the references, there is an arXiv source with official publication.
    * Simple and scalable predictive uncertainty estimation using deep ensembles: NIPS 2017

**Questions:**

1. Why is Deep Sea (deterministic) considered a suitable environment for justifying E-MCTS? Does Deep Sea have any variation in state transitions or rewards that can induce epistemic uncertainty? In a deterministic environment, once we observe a sample, there is only one possible outcome, and further interactions do not reduce uncertainty by obtaining more samples of the same state-action pair.
2. Will the new exploration bonus $\beta\sqrt{\sigma^2_q(\hat{s}_k,a)}$ eventually converge to zero?

**Details Of Ethics Concerns:**

This paper has been accepted by EWRL (European Workshop on Reinforcement Learning) 2023. While it appears that there is no official proceedings for EWRL 2023, the workshop utilizes openreview and publishes all accepted papers, creating an automatic proceeding (https://openreview.net/forum?id=w4JFRTD0_R4#). I would like to kindly request the AC to double-check this information.

---

> ### Author Response · Authors · 2023-11-19
>
> Dear reviewer,
>
> thank you for the detailed review and the great comments, notes and questions.
>
> 1) In our understanding, Alpha/MuZero are designed for and traditionally evaluated in deterministic environments where exploration may still be important (Atari hard exploration, Go, Chess), which we believe makes deterministic Deep Sea a suitable testbed for the hard-exploration deterministic-environment setting. Note that the lack of information sourced in unobserved transitions in enough by itself to introduce epistemic uncertainty which is important to take into account and propagate in the planning tree, even in the presence of entirely deterministic transitions and rewards.
> Since stochastic Deep Sea has stochastic transitions, we do not believe it is best suited to evaluate E-MCTS which we design on the framework of baseline Alpha/MuZero for a determinstic transition environments.
> On the other hand, stochastic-reward deterministic-transition Deep Sea (perhaps not the classic variation of the environment or a classic Alpha/MuZero setting, but also not an unreasonable one) can be used to evaluate E-MCTS' in the non-deterministic reward setting. We will make sure the above is discussed in the paper.
>
> 2) We believe this depends on the choice of transition-epistemic-uncertainty estimation mechanism.
> For example, if we choose 0 uncertainty for visited transitions and 1 for unvisited (suitable for a deterministic environment with bounded rewards), $ \sigma_{q}(s,a) $ will be zero once all transitions in all trajectories evaluated from this node have been visited.
> However, if we choose $ \sigma_r = \frac{1}{\epsilon + \sqrt{N}} > 0 $ which is suitable for a reward function $ r(s,a) \sim N(\mu(s,a), 1) $, the epistemic uncertainty $\sigma_r, \sigma_q $ will not shrink to exactly zero except in the limit of infinite visitations.
> In Figure 3 we demonstrate that with RND the uncertainty goes numerically to zero in practice.
>
> Does this address the reviewer's questions?

---

> > ### Comment · Reviewer_9sR2 · 2023-11-20
> >
> > Thank you for your response.
> >
> > To our knowledge, a major source of complexity in zero-like algorithms arises from opponent policies. Although deterministic environments also introduce epistemic uncertainty in model-based RL, we typically face additional sources of uncertainty when adapting MCTS for complex problems. We suggest using Deep Sea without a search (i.e., directly using the value or policy networks) to assess epistemic uncertainty in the future. You can also conduct MCTS with uncertainty in problems that typically utilize MCTS, such as two-player zero-sum games, as you plan to do in your future submission. Otherwise, we anticipate that methods similar to Go-Explore could easily solve Deep Sea but might face challenges in more complex experiments (even with a 50x50 size, there are only 2,500 states).
> >
> > We believe it would be beneficial to discuss the topic of exploration bonus convergence in your future submission. For instance, RND's discussion of the noisy-TV problem provides valuable insights into the convergence or limitations.
> >
> > This is indeed a promising topic, but it requires further evidence to validate E-MCTS in more complex environments where the use of MCTS is justified. This would help confirm the effectiveness of the improved version of MCTS.
> >
> > I have increased the Confidence level to 5.
> > If experiments with Stochastic Deep Sea show promising results, I can raise the Rating to 6.
> > If experiments with two-player zero-sum games (which seem to require more time) are successful, I can raise the Rating to 8.
> > Otherwise, I look forward to seeing good results in your future submissions.

---

### Official Review · Reviewer_qzYy · 2023-11-02

**Soundness:** 1 poor
**Presentation:** 2 fair
**Contribution:** 2 fair
**Rating:** 3
**Confidence:** 3

**Summary:**

The paper presents Epistemic-MCTS (E-MCTS), an advancement of the Monte-Carlo Tree Search (MCTS) algorithm in deep model-based reinforcement learning, targeting improved exploration. By incorporating and disseminating epistemic uncertainty within MCTS, enhanced exploration strategies emerge. The approach employs an RND network as proxy-measure of novelty and calculate variance estimates of unobserved states, which is subsequently propagated through the search tree to guide exploration during planning. It is tested against three baselines on the Deep Sea benchmark and outperforms the baselines, though the baselines gradually approach its performance.

**Strengths:**

S1: Proposes a new exploration strategy for planning using MCTS based on a proxy measure of novelty, i.e., RND, and provides a practical algorithm that performs well in the Deep Sea domain with minimal computational overhead (though it varies based on the network architecture size).

S2: The method offers a mathematical approach to propagate uncertainty in predictions throughout the planning process.

**Weaknesses:**

The key idea appears to be allowing the search to recognize the uncertainty in the value predicted by the learned model for unobserved states and directing the search towards actions with greater uncertainty (higher variance). Additionally, a proxy-measure of novelty is employed to estimate this uncertainty.

W1: Some key concepts in the realm of exploration in MCTS haven't been touched upon. While the visitation counts themselves represent the uncertainty in the Q-value estimate at a node, other researchers have utilized the variance of predicted Q-value [1,2] and maintained it at each tree node with a Gaussian distribution to guide exploration during action selection. [3] adopts a more systematic approach to measure uncertainty in the Q-value of the unobserved state using a Gaussian process, promoting exploration based on the upper confidence derived from the variance of the GP. The advantage of using a proxy-measure of novelty over these methods isn't evident.

W2: The experiment section is somewhat limited in the diversity of the problems, making it challenging to deem the approach as robust and significant. While Deep Sea may be an illustrative example to showcase the strengths of E-MCTS, a broader experimental setting is essential to validate its edge over established methods.

W3: The writing could benefit from some refinement. For instance, the context in which "epistemic uncertainty" was introduced remained unclear until section 3. Moreover, by referencing the AlphaZero and MuZero models, it seems the authors might be differentiating between whether the transition model is learned or provided as a simulator. However, the current phrasing is somewhat confusing.

[1] Tesauro, Gerald, V. T. Rajan, and Richard Segal. 2012. “Bayesian Inference in Monte-Carlo Tree Search.” arXiv [Cs.LG]. arXiv. http://arxiv.org/abs/1203.3519.

[2] Bai, Aijun, Feng Wu, and Xiaoping Chen. 2013. “Bayesian Mixture Modelling and Inference Based Thompson Sampling in Monte-Carlo Tree Search.” Advances in Neural Information Processing Systems 26: 1646–54.

[3] Mern, John, Anil Yildiz, Zachary Sunberg, Tapan Mukerji, and Mykel J. Kochenderfer. 2020. “Bayesian Optimized Monte Carlo Planning.” arXiv [Cs.AI]. arXiv. http://arxiv.org/abs/2010.03597.

**Questions:**

Q1: How does the proposed method compare to the methods mentioned in W1? In what aspects is it better?

Q2: Can you present results from broader and more realistic experimental settings, such as Procgen?

Q3: In Figure 2 (right), the average regret continues to decrease even with high values of Beta. This trend seems counter-intuitive, implying that optimal regret is achieved mainly through high exploration and minimal exploitation. Could you elaborate on this observation?

---

> ### Author Response · Authors · 2023-11-19
>
> Dear reviewer,
>
> thank you for the detailed review and the excellent comments, questions and additional references.
>
> 1. The methods in W1 and EMCTS are designed to contend with different settings and different sources of uncertainty, in our understanding.
> E-MCTS addresses the case where the model used in MCTS is itself uncertain (i.e. there is uncertainty about the means of the transition, reward and value functions), while the methods in W1 address other cases (where the environment and thus the model is a POMDP or where the leaves in the tree have predictions with arbitrary variances but still true means with respect to the environment).
> We believe that is also E-MCTS's main advantage and source of novelty.
>
> 2. We are in the process of evaluating EMCTS in  two player zero-sum games but were not able to provide results within the timeframe of this rebuttal.
>
> 3. We first note that Figure 2 contains evaluation episodes' regret, not training episodes' regret. We will make sure this is clear in the paper.
> Due to the off policy training of the agents with Reanalyze and the properties of Deep Sea, the agents learn to solve the problem essentially immediately upon first interaction with the goal.
> As a result, the task becomes exclusively a "find the goal" task, and if the agents' off-policy training is stable no training episode will influence the evaluated regret after finding the goal for the first time.
>
> Does this address the reviewer's questions?

---

### Official Review · Reviewer_sVMG · 2023-11-04

**Soundness:** 3 good
**Presentation:** 3 good
**Contribution:** 2 fair
**Rating:** 5
**Confidence:** 3

**Summary:**

Monte Carlo Tree Search (MCTS) as used in AlphaZero and MuZero implicitly reduces aleatoric uncertainty through its rollouts, but does not traditionally account for epistemic uncertainty, which can hinder exploration/

This paper introduces Epistemic MCTS (E-MCTS), that extends standard MCTS to utilize epistemic uncertainty (uncertainty in estimates that are reducible with more observations) to guide rollouts and decisions, which benefits exploration. E-MCTS allows the agent to pe

E-MCTS is tested on a variety of configurations of the DeepSea environment in the Behavior Suite, which is a hard exploration task.
They find that their variant of E-MCTS is able to achieve a high return faster than baselines, and is able to discover states more quickly than the baselines. Additionally, they find that the benefits (in terms of regret) of E-MCTS scale with the environment size.

**Strengths:**

The motivation for E-MCTS is well-articulated, and the papers makes it clear what gap E-MCTS intends to fill.

The method itself appears to be sound, makes algorithmic sense, and seems to also be a good solution to the identified problem.

The presentation of the paper is good: it is well-written.

The paper convincingly shows the effectiveness of their method on a deep exploration task.

**Weaknesses:**

My primary issues with the paper have to do with experimentation. The paper does not test on a diverse set of environments and rather tests on different configurations (albeit with different difficulties) of the same DeepSea environment. While the DeepSea environment is certainly nontrivial and challenging, it has a very specific reward structure for which we would expect E-MCTS to perform well. While it serves as an excellent demonstration of the potential benefits of the proposed method, it does not demonstrate more generally the ability and tradeoffs of the proposed method. For example, does the introduction of the utilization of epistemic uncertainty estimates adversely impact results in environments where deep exploration is not required?  Moreover, to my knowledge, the proposed method is not compared against other domains for which MCTS is typically used. It would be nice to test E-MCTS on environments where MuZero is applied (albeit in a tractable way).


The paper has a lot of merits, and I believe with more comprehensicve experimentation it may warrant acceptance. Even results on a diverse set of standard environments, especially ones where MCTS is typically applied, would greatly improve the paper. As it stands, my interpretation of the experiments is that they "demonstrate" the potential of the method, but do not show the "effectiveness", which can be shown with other domains. Even showing that E-MCTS works well or does not harm MCTS in standard environments will show that its potential is not limited to environments necessitating deep exploration.


Suggestions:
- Given that DeepSea is the only environment tested, would recommend writing some description of the DeepSea environment.


Nits/typos:
- Section 2.1 "the the" -> "the"
- Figure 2 Caption: "perparameter" -> "hyperparameter"
- Table 3 in appendix has 'self-play networking updating inerval'. The "inerval" should be "interval"

**Questions:**

1. In figure 2, why 3 seeds for the 30x30 domain on the right, but 5 seeds on the left (where presumably the 30x30 domain is ran?)
2. How might exploration be balanced, tuned, or annealed over time in environments where deep exploration is not required?

---

> ### Author Response · Authors · 2023-11-19
>
> Dear reviewer,
>
> thank you for the detailed review and the excellent comments and questions.
>
> 1. Figure 2: left compares different agents across domains, from 10 by 10 to 50 by 50 (Deep Sea size), while Figure 2: right is just 30 by 30.
> We deemed the reduction of the standard deviation in the left figure as more important than the right, where our objective was to support the hypothesis that from a certain value of $\beta$ onward, a large range of values were sufficient.
> We note that each point in Figure 2: right is composed of 3 seeds, for a total of 24 independent seeds in the figure.
>
> 2. In the simple case where exploration is actually detrimental to learning, $\beta$ could simply be turned to zero of course. If the value and uncertainty are tuned between 0,1 (reasonable in Alpha/MuZero), and in challenging domains where exploration might be benefited from but it's not clear what is the correct exploration-exploitation tradeoff (for example Chess or Go), we believe that an exploration schedule that runs multiple episodes in parallel (or alternating) with different values of $\beta$ (let's say, one large one small), in addition to random exploration episodes as well as deterministic exploitation episodes that rely on policy / value churn for exploration, will result in a very strong exploration heuristic, although in the paper we only evaluated the very edge of this idea by alternating E-MCTS exploration and deterministic exploitation episodes. It is of course also possible to use a decay schdule for $\beta$, although we believe in many environments the natural decay of uncertainty might be sufficient.

---

### Author Response · Authors · 2023-11-19

Dear reviewers,

Thank you for your valuable time, detailed reviews and many insightful comments, questions and suggestions.

We were hoping to be able to present results on one of the popular 2-player zero-sum games to empirically evaluate E-MCTS in more general settings than simple hard-exploration problems but unfortunately we weren't able to within the time frame of this rebuttal.
Seperately, In the time between the submission and the rebuttal we have also made great strides in theoretical analysis which result among others in many changes to presentation and problem formulation.

We believe that rather than uploading as part of this rebuttal an almost new paper in terms of presentation without additional results, the more appropriate choice would be to withdraw this submission in favor of a complete submission with improved presentation and additional results in the future.

We would like to thank the reviewers again for their insightful comments, which will further improve the presentation of the next iteration of this paper.